# MedSelect: Selective Labeling for Medical Image Classification Using Meta-Learning

**Damir Vrabac**[*1]                                                           DVRABAC@CS.STANFORD.EDU
[1] *Department of Computer Science, Stanford University, Stanford, CA, USA*
**Akshay Smit**[*1]                                                           AKSHAYSM@CS.STANFORD.EDU
**Yujie He**[*1]                                                              YUJIEHE@CS.STANFORD.EDU
**Andrew Y. Ng**[1]                                                           ANG@CS.STANFORD.EDU
**Andrew L. Beam**[2]                                                         ANDREW_BEAM@HMS.HARVARD.EDU
[2] *Department of Epidemiology, Harvard T.H. Chan School of Public Health, Boston, MA, USA*

**Pranav Rajpurkar**[3]                                                       PRANAV_RAJPURKAR@HMS.HARVARD.EDU
[3] *Department of Biomedical Informatics, Harvard Medical School, Boston, MA, USA*

**Editors:** Under Review for MIDL 2022

## Abstract

We propose a selective labeling method using meta-learning for medical image interpretation in the setting of limited labeling resources. Our method, MedSelect, consists of a trainable deep learning model that uses image embeddings to select images to label, and a non-parametric classifier that uses cosine similarity to classify unseen images. We demonstrate that MedSelect learns an effective selection strategy outperforming baseline selection strategies across seen and unseen medical conditions for chest X-ray interpretation. We also perform an analysis of the selections performed by MedSelect comparing the distribution of latent embeddings and clinical features, and find significant differences compared to the strongest performing baseline. Our method is broadly applicable across medical imaging tasks where labels are expensive to acquire.

**Keywords:** Active learning, Meta-learning, Selective labeling, Chest X-ray interpretation

## 1. Introduction

Large labeled datasets have enabled the application of deep learning methods to achieve expert-level performance on medical image interpretation tasks (Topol, 2019; Litjens et al., 2017). However, expert labeling is expensive at scale and approaches to learn in the presence of limited labeled data are improving (Gadgil et al., 2021; Vu et al., 2021). Active learning is an approach for reducing the amount of labeled data needed by having an algorithm select a subset of images that should be labeled (Cohn et al., 1996; Liu, 2004; Hoi et al., 2006). While active learning strategies can be designed to iteratively select examples to label over several steps (Azimi et al., 2012; Guo and Schuurmans, 2008), a *selective labeling* strategy to select examples in a single step may be more useful and practical for existing medical image labeling pipelines (Gu et al., 2012; Yang and Loog, 2018).

In this work, we use meta-learning to learn a selective labeling strategy on medical images. Learning selective labeling strategies (and more generally active learning strategies) end-to-end via meta-learning represents an advance over using task-specific heuristics for

---

[*] Contributed equally

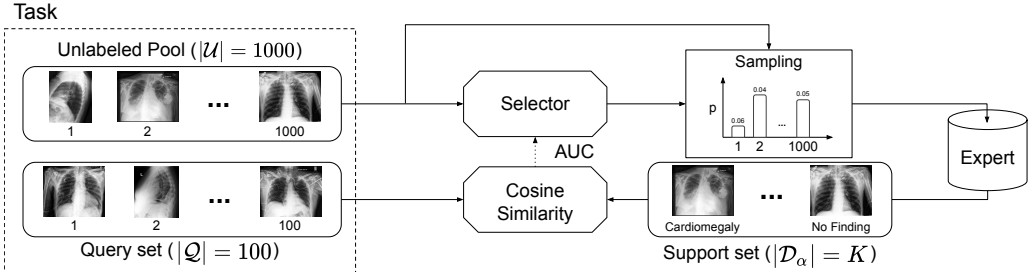

Figure 1: MedSelect method. The unlabeled pool of chest X-ray embeddings $\mathcal{U}$ is passed to the selector. The selector outputs a probability distribution over the examples in $\mathcal{U}$ which we sample $K$ examples from. The sampled chest X-ray embeddings are labeled by 'the Expert', creating the support set $\mathcal{D}_\alpha$. This set, $\mathcal{D}_\alpha$, is used to fit a cosine similarity classifier which is evaluated on the query set $\mathcal{Q}$ based on the AUROC score which is use as the reward function for the selector.

selecting instances to label (Gal et al., 2017; Joshi et al., 2009), and has been shown to be effective for simple natural image classification tasks (Bachman et al., 2017; Woodward and Finn, 2017; Mas et al., 2019), including (Contardo et al., 2017) which is most similar to our setup. However, their development and effectiveness on more complex, real-world medical image classification settings remains unexplored.

In particular, we develop *MedSelect*, a deep-learning based selective labeling method for medical images. MedSelect consists of a trainable selector that selects medical images using image embeddings obtained from contrastive pretraining, and a non-parametric classifier that classifies unseen images using cosine similarity. MedSelect is trained end-to-end with the combination of backpropagation and policy gradients. On chest X-ray interpretation tasks, we demonstrate that MedSelect outperforms a random selection strategy, a selection strategy based on clustering of image embeddings, and a deep learning based selection strategy using clinical data. Furthermore, we demonstrate that MedSelect generalizes to different medical conditions, even to ones unseen during meta-training. We also perform an analysis of the selections performed by MedSelect comparing latent embeddings and clincal features. We find that MedSelect tends to select X-rays that are closer together in a latent embedding space compared to other strategies. It almost exclusively selects frontal X-rays and tends to select X-rays from younger patients. The distribution of sex in X-rays sampled by MedSelect tends to be more balanced than for other strategies.

## 2. Data

**Labeled Chest X-rays.** We make use of the CheXpert dataset, containing 224,316 chest X-rays labeled for several common medical conditions (Irvin et al., 2019). We randomly sample 70% of the dataset to use for meta-training, 15% for meta-validation and 15% for meta-testing with no patient overlap. We only make use of medical conditions with a 5% or greater prevalence of positive labels, which are: Consolidation, Enlarged Cardiomedi-

astinum, Cardiomegaly, Pneumothorax, Atelectasis, Edema, Pleural Effusion, and Lung Opacity. We do not use rarer conditions.

Given an X-ray from the CheXpert dataset, each of the medical conditions can be labeled as "uncertain", "positive" or "negative". If an X-ray contains no abnormalities, it is labeled as "No Finding" The ground-truth labels for these X-rays are produced by the CheXpert labeler, which is an automatic radiology report labeler. It extracts labels for each X-ray study using the corresponding free-text radiology report which describes the key observations in the X-ray.

**Task Construction.** In our meta learning setting, each task $\mathcal{T} = (c, \mathcal{U}, \mathcal{Q})$ is a tuple consisting of a medical condition $c$, a set $\mathcal{U}$ of 1000 unlabeled chest X-rays which we refer to as the *unlabeled pool*, and a set $\mathcal{Q}$ of 100 labeled chest X-rays which we refer to as the *query set*. The medical condition $c$ is sampled randomly among the conditions we use. We consider the setting where 50% of the X-rays in $\mathcal{U}$ and $\mathcal{Q}$ are sampled so that they are labeled positive for the sampled condition $c$. The other 50% of X-rays are sampled so that they are labeled as No Finding, i.e. no abnormalities are found in these X-rays. There is no overlap between the X-rays in $\mathcal{Q}$ and $\mathcal{U}$. The corresponding binary label for an X-ray is positive if condition $c$ is found in the X-ray, else it corresponds to "No Finding". Each task is thus a binary classification problem, in which an X-ray must be classified as either positive for the condition $c$, or as No Finding. We produce 10,000 tasks for meta-training, 1000 tasks for meta-validation, and 2000 tasks for meta-testing. There are no tasks with duplicate patients. Since we only use 8 medical conditions from the CheXpert dataset, each particular medical condition (e.g. Edema) occurs in several tasks, but the X-rays for each task are randomly sampled. For each task we sample 1100 X-rays, since $|\mathcal{U}| = 1000$ and $|\mathcal{Q}| = 100$. When constructing a meta-training task (resp. meta-validation, meta-testing) we randomly sample the X-rays in $\mathcal{Q}$ and $\mathcal{U}$ from the 70% split of the CheXpert dataset used for meta-training (resp. 15% for meta-validation, 15% for meta-testing). There is no overlap between the X-rays used for meta-training, meta-validation and meta-testing.

We randomly selected two medical conditions, Edema and Atelectasis, to be held out during meta-training so that the generalization performance of our models can be evaluated on conditions unseen during meta-training. We refer to Edema and Atelectasis as the *holdout conditions*, whereas all other conditions are referred to as *non-holdout conditions*. For the meta-training and meta-validation tasks, we only sample $c$ from the non-holdout conditions. Half of the meta-testing tasks use the non-holdout conditions, while the other half use the holdout conditions. During meta-testing, we evaluate the performance of our models on both non-holdout and holdout conditions. There is no overlap between the X-rays used for holdout conditions and non-holdout conditions.

## 3. Methods

MedSelect consists of a trainable selector and a non-parametric classifier. The selector consists of a bidirectional LSTM that selects medical images using image embeddings obtained from contrastive pretraining. The classifier uses cosine similarity to classify unseen images. Our approach is illustrated in Figure 1.

The selector takes in a pool of unlabeled medical image embeddings and outputs a probability distribution over the unlabeled images. We use the probability distribution

to sample a small set of images, for which we obtain labels. In our setup, the labels are provided by the automated CheXpert labeler, which serves as a strong proxy for the expert (Irvin et al., 2019). The sampled images and their labels constitute the *support set* which is used to fit the cosine similarity classifier before the classifier is used to infer on the query set.

## 3.1. Selector

We use a single-layer bidirectional LSTM (BiLSTM) (Hochreiter and Schmidhuber, 1997; Gers et al., 1999) with 256 hidden units as the selector model. We obtain image embeddings for each X-ray from a ResNet-18 (He et al., 2015) pretrained by (Sowrirajan et al., 2020) on the CheXpert dataset using Momentum Contrast (MoCo) pretraining. We only use embeddings from this ResNet-18, and do not modify its parameters. The unlabeled chest X-ray embeddings in $\mathcal{U} = \{x^{(i)}\}_{i=1}^{1000}$ are fed as input to the BiLSTM. The resulting outputs are normalized and treated as a multinomial probability distribution over the examples in $\mathcal{U}$. We denote the parameters of the BiLSTM by $\theta$, and the multinomial probability distribution is denoted by $\mathbb{P}_\theta(\alpha|\mathcal{U})$, where $\alpha \in \{0, 1\}^{1000}$ and $\alpha_i = 1$ indicates that the X-ray $x_i$ is selected to be labeled. Since we only select $K$ X-rays from $\mathcal{U}$ for labeling, we have $\sum_{i=1}^{1000} \alpha_i = K$. Note that the selector has no access to the labels and can be used in the same way on the holdout conditions as in the non-holdout conditions.

## 3.2. Classifier

We use a non-parametric chest X-ray classification model as the classifier. (Sowrirajan et al., 2020) showed that training a linear head on top of the frozen MoCo-pretrained ResNet-18 achieved superior performance on CheXpert, outperforming ImageNet pretraining. Furthermore, the MoCo-pretrained embeddings did not require any labeled data. Our classifier, which performs cosine similarity on the MoCo-pretrained image embeddings, does not have trainable parameters, reducing computational requirements and simplifying the training procedure.

Given the unlabeled pool $\mathcal{U}$, the selector model selects a set of $K$ examples from $\mathcal{U}$ to retrieve labels from the expert. We denote the set of these selected examples by $\mathcal{D}_\alpha$. Note that $|\mathcal{D}_\alpha| = K$. The classifier is given the set $\mathcal{D}_\alpha$ along with the corresponding labels. We denote the examples in $\mathcal{D}_\alpha$ by $(x_\alpha^{(i)})_{i=1}^K$ with corresponding labels $(y_\alpha^{(i)})_{i=1}^K$. The label $y_\alpha^{(i)} = 1$ if the example $x_\alpha^{(i)}$ is positive for the condition $c$, otherwise $y_\alpha^{(i)} = 0$ which implies that the example $x_\alpha^{(i)}$ is labeled as No Finding.

The classifier then computes two averages: $p = \frac{\sum_{i=1}^K \mathbf{1}\{y_\alpha^{(i)}=1\}x_\alpha^{(i)}}{\sum_{i=1}^K \mathbf{1}\{y_\alpha^{(i)}=1\}}$ and $n = \frac{\sum_{i=1}^K \mathbf{1}\{y_\alpha^{(i)}=0\}x_\alpha^{(i)}}{\sum_{i=1}^K \mathbf{1}\{y_\alpha^{(i)}=0\}}$. $p$ is simply the average of all examples in $\mathcal{D}_\alpha$ that are positive for condition $c$, and $p$ can be considered a prototypical vector for such examples. Similarly, $n$ is the average of all examples in $\mathcal{D}_\alpha$ that are labeled as No Finding.

Given a new example $x^{\mathcal{Q}}$ from $\mathcal{Q}$ we compute the difference of the cosine similarities:

$$\frac{x^{\mathcal{Q}} \cdot p}{\|x^{\mathcal{Q}}\|\|p\|} - \frac{x^{\mathcal{Q}} \cdot n}{\|x^{\mathcal{Q}}\|\|n\|}.$$

Our experimental methodology is shown in Figure 1.

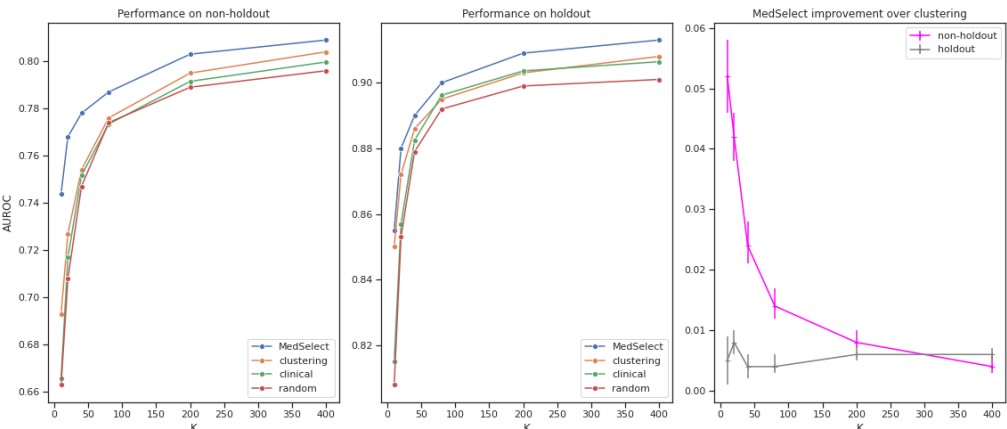

Figure 2: **Performance of MedSelect and the baseline selectors.** Left: AUROC averaged over all non-holdout conditions. Middle: AUROC averaged over all holdout conditions. Right: Improvements in AUROC obtained by MedSelect over the clustering baseline with 95% confidence intervals, averaged over holdout and non-holdout conditions. We show the results for $K = 10,\ 20,\ 40,\ 80,\ 200,\ 400$.

### 3.3. Optimization

Given the classifier's outputs for each X-ray in the query set $\mathcal{Q}$, we compute the AUROC score using the corresponding labels. We denote this AUROC score by $\mathcal{R}\left(\mathcal{D}_\alpha, \mathcal{Q}\right)$, and this serves as the reward that we wish to optimize. We define our objective function as follows, which we wish to maximize:

$$\mathcal{L} = \mathbb{E}_{\alpha \sim \mathbb{P}_\theta(\alpha|\mathcal{U})}\left[\mathcal{R}\left(\mathcal{D}_\alpha, \mathcal{Q}\right)\right]$$

We make use of the policy-gradient method (Williams, 1992) to optimize this objective and approximate the expected value $\mathbb{E}_{\alpha \sim \mathbb{P}_\theta(\alpha|\mathcal{U})}$ by a single Monte-Carlo sample. Through empirical experiments, we find that the training stability improves if we subtract a baseline reward from $\mathcal{R}\left(\mathcal{D}_\alpha, \mathcal{Q}\right)$. Specifically, let the reward obtained using the random baseline be $b\left(\mathcal{U}, \mathcal{Q}\right)$. Then the final gradient which we use for gradient ascent is:

$$\nabla_\theta \mathcal{L} = \mathbb{E}_{\alpha \sim \mathbb{P}_\theta(\alpha|\mathcal{U})}[(\mathcal{R}\left(\mathcal{D}_\alpha, \mathcal{Q}\right) \\ - b\left(\mathcal{U}, \mathcal{Q}\right))\nabla_\theta \log \mathbb{P}_\theta\left(\alpha|\mathcal{U}\right)]$$

**Training details** We use the Adam optimizer (Kingma and Ba, 2017) with a learning rate of $10^{-4}$ and a batch size of 64 tasks. We train all our models for 5 epochs. We use a single GeForce GTX 1070 GPU with 8 GB memory. During meta-training, we periodically evaluate our deep learning models on the meta-validation set and save the checkpoint with the highest meta-validation performance.

### 3.4. Baseline Comparisons

We compare MedSelect to three baseline strategies: random, clustering, and clinical.

**Random.** We implement a selector that randomly selects $K$ examples without replacement from the pool of unlabeled data $\mathcal{U}$. This selector provides an expected lower bound of the performance compared to more sophisticated selection strategies. We refer to this strategy as the *random baseline*.

**Clustering.** Another non-parametric baseline is a K-Medoids clustering method where the data points in $\mathcal{U}$ are partitioned into $K$ clusters. Unlike K-Means, the centroids obtained from K-Medoids are datapoints in $\mathcal{U}$. The centroid of each cluster is selected for labeling, and is passed along with the label to the classifier. We refer to this method as the *clustering baseline* and expect it to be a stronger baseline than the random baseline.

**Clinical.** We also implement a similar BiLSTM selector as in MedSelect that only takes three features as input, namely age, sex, and laterality where the laterality indicates whether it is a frontal or lateral chest X-ray. Thus, the input to this selector consists of three dimensional vectors for each corresponding chest X-ray and the selector has no pixel-level information of the chest X-rays. We refer to this strategy as the *clinical baseline*.

## 4. Results

### 4.1. Can MedSelect learn a selective labeling strategy that outperforms baseline comparisons?

We find that MedSelect is able to significantly outperform random and clustering based selection strategies, as well as a strategy based on clinical metadata. The clustering baseline outperforms the random and clinical baselines, especially for small $K$. For $K = 10$ on non-holdout conditions, clustering achieves 0.693 AUROC compared to 0.663 for random and 0.666 for clinical. For $K = 10$ on holdout conditions, clustering achieves 0.850 AUROC compared to 0.808 for random and 0.815 for clinical. We show the AUROC scores obtained by MedSelect and the baseline selectors, as well as the improvements obtained by MedSelect over the clustering baseline, in Figure 2.

We find that MedSelect achieves the highest performance compared to the baselines on both the non-holdout as well as the holdout conditions. On the non-holdout conditions, MedSelect obtains statistically significant improvements over the clustering baseline for all values of $K$, with the maximum improvement of 0.052 (95% CI 0.046, 0.058) obtained for $K = 10$.

On the holdout conditions, MedSelect obtains statistically significant improvements over the clustering baseline for all $K$. The largest improvement of 0.008 (95% CI 0.006, 0.010) is obtained for $K = 20$. However, the improvements are smaller than for the non-holdout conditions. This may be because the clustering baseline is non-parametric and does not require learning, whereas the parameters of the BiLSTM were specifically optimized using the non-holdout conditions.

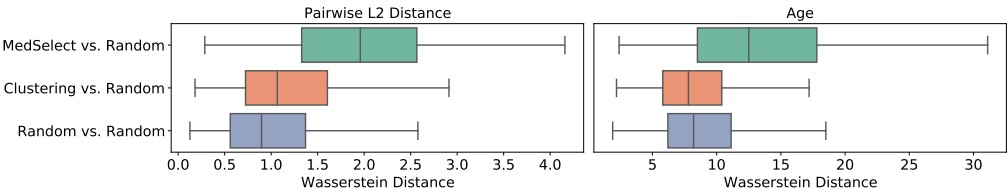

Figure 3: Left: Wasserstein distances corresponding to pairwise L2 distances between selected X-rays. Right: Wasserstein distances corresponding to age of the selected X-rays. For comparison, we also compute the Wasserstein distance between two random selectors with different random seeds.

### 4.2. How do the X-rays selected by MedSelect differ from those selected by the baselines?

We first compare the distributions of age, sex, and laterality of X-rays selected by MedSelect and the clustering baseline. We set $K = 10$ and test both selectors with non-holdout conditions. For each meta-testing task, we compute the average age, percentage female and percentage frontal X-rays selected by the two selectors, and test the difference of means of these statistics over meta-testing tasks using independent two-sample t-tests.

We find that 99.96% of the samples selected by MedSelect are frontal X-rays while only 77.58% selected by the clustering baseline are frontal ($p < 0.001$). The percentage female selected by MedSelect and the clustering baseline are on average 42.31% and 39.46%, respectively ($p < 0.001$). The mean ages selected by MedSelect and the clustering baseline are on average 45.2 and 54.8, respectively ($p < 0.001$). Thus, we observe that MedSelect selects (1) mostly frontal X-rays, (2) more evenly from both sexes, and (3) younger samples.

We first hypothesize that MedSelect may be selecting X-rays that are further away from each other in the embedding space compared to the clustering baseline. To test this hypothesis, for each meta-testing task, we compute the pairwise L2 distances between embeddings of X-rays selected by MedSelect, then compute the mean and maximum of the pairwise distances. We also do this for the clustering baseline. We then calculate the p-values comparing the mean of these statistics over meta-testing tasks for MedSelect vs. the clustering baseline, using independent two-sample t-tests.

We find that the mean pairwise distances between embeddings of X-rays selected by MedSelect are 7.76, while those between embeddings of X-rays selected by the clustering baseline are 8.78 ($p < 0.001$). Similarly, the maximum pairwise distance between embeddings of X-rays selected by MedSelect is 11.06, while that between embeddings of X-rays selected by the clustering baseline is 12.37 ($p < 0.001$). We also find that the mean pairwise distances between embeddings of frontal X-rays selected by MedSelect is 7.76, while that between embeddings of frontal X-rays selected by the clustering selector is 8.22 ($p < 0.001$). Similarly, the maximum pairwise distances between embeddings of frontal X-rays selected by MedSelect is 11.06, while those between embeddings of frontal X-rays selected by the clustering selector is 11.69 ($p < 0.001$). Therefore, we find the opposite of our hypothesis to be true: MedSelect selects X-rays that are closer to each other in the embedding space compared to the clustering baseline.

Second, we hypothesize that the difference in empirical distribution of ages of X-rays between MedSelect and the random baseline is significantly higher than that between clustering and random. We use the Wasserstein distance to investigate the difference in the distributions of X-rays selected by different selectors. The Wasserstein distance of two distributions is the L1 distance between the quantile functions of these distributions (Ramdas et al., 2015). For a meta-testing task $\mathcal{T}_i$, let $\hat{\mu}_X^i$ be the empirical distribution of the age for X-rays selected by MedSelect. We similarly compute $\hat{\mu}_C^i$ for the clustering baseline and $\hat{\mu}_R^i$ for the random baseline. We compute $d\left(\hat{\mu}_X^i, \hat{\mu}_R^i\right)$, the Wasserstein distance between $\hat{\mu}_X^i$ and $\hat{\mu}_R^i$, for all $i$ and show these Wasserstein distances in Figure 3 on the right. We also show $d\left(\hat{\mu}_C^i, \hat{\mu}_R^i\right)$. We find that the average Wasserstein distance between MedSelect and the random baseline is 13.58, larger than that between the clustering baseline and the random baseline which is 8.43 ($p < 0.001$). Thus we find our hypothesis to be true: the difference in empirical distribution of ages of X-rays between MedSelect and the random baseline is significantly higher than that between clustering and random.

Finally, we hypothesize that the above difference in the empirical distributions for ages also holds for pairwise L2 distances between X-ray embeddings. We repeat the above procedure while replacing ages of the X-rays with the L2 distances between the embeddings of each pair of selected X-rays. For a meta-testing task $\mathcal{T}_i$, we compute the L2 distances between each pair of X-ray embeddings selected by MedSelect. Let $\hat{\nu}_X^i$ be the empirical distribution over these pairwise distances. We similarly define $\hat{\nu}_C^i$ for the clustering baseline and $\hat{\nu}_R^i$ for the random baseline. We compute $d\left(\hat{\nu}_X^i, \hat{\nu}_R^i\right)$ and $d\left(\hat{\nu}_C^i, \hat{\nu}_R^i\right)$ for all $i$ and show these in Figure 3 on the left. We find that the average Wasserstein distance between MedSelect and the random baseline is 2.01, larger than that between the clustering baseline and random baseline which is 1.22 ($p < 0.001$). This confirms our hypothesis that the difference in the empirical distributions between MedSelect and the random baseline is significantly higher than that between clustering and random.

## 5. Conclusion

In this work, we present MedSelect, a selective labeling method using meta-learning for medical image interpretation. MedSelect significantly outperforms random and clustering based selection strategies, as well as a heuristic strategy based on clinical metadata. MedSelect successfully generalizes to unseen medical conditions, outperforming other strategies including clustering with statistical significance.

Our study has three main limitations. First, we simplify the multi-label chest X-ray interpretation problem to several binary classification tasks. Second, the expert in our method consists of a rule-based labeler that uses the corresponding radiology report to label each chest X-ray, and future work should measure the cost-performance tradeoffs for human annotation. Third, our experiments considers an unlabeled pool of only 1000 examples and future work should extend this approach for larger pool sizes.

We expect our approach to be broadly useful in the medical domain beyond chest X-ray labeling. In real world clinical settings where unlabeled medical data is abundant but expert annotations are limited, MedSelect would facilitate the development of classification models and improve model training through selective labeling of a limited number of medical images.

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
