# OpenReview forum: "MedSelect: Selective Labeling for Medical Image Classification Using Meta-Learning"
_MIDL.io/2022/Conference — MIDL 2022_

### Official Review · Reviewer_FBj5 · 2022-01-23

**Confidence:** 3
**Preliminary Rating:** 3
**Recommendation:** Poster

**Summary:**

The paper proposes a meta-learning-based method for selective labeling (active learning).
The method creates 10K tasks where each task consists of 1000 unlabeled images and 100 labeled images for the query set.
Images from the unlabeled sets are fed to a pre-trained MoCo network to obtain latent representations which then be given to a selector BiLSTM network to obtain the probability of each sample to determine whether the sample should be labeled or not.
K samples with the highest probability are then labeled and used to train a classifier which is meta tested in the query set.
During optimization, the classification loss is minimized in all tasks w.r.t. the parameters of BiLSTM.
Experiments are performed on CheXpert dataset and the results are compared with clustering-based selection, random selection and a modified version of the proposed method where the selector network takes some clinical features instead of MoCo embeddings.
The results demonstrate that the proposed method improves existing methods.



**Strengths:**

- The idea of using meta-learning for selective labeling is novel and quite interesting.
- The improvement achieved compared to other methods, especially the clustering-based one, is quite significant.

**Weaknesses:**

1 - Experiments are limited to show the performance of the proposed method:
- It would be useful to perform experiments on multiple datasets to demonstrate that the method extends to different datasets.
- There are only 2 unseen conditions that are used as held-out. Do the results hold for different unseen conditions? More experiments can be performed where Edema and Atelectasis conditions are moved to training and 2 conditions from training are used as held-out.

2 - Comparison with existing methods: Currently there are only comparisons with the clustering-based method as a strong baseline. Comparison with other baselines (e.g. the ones implemented in DeepAL [1])would strengthen the paper in terms of positioning the method in the active learning literature.

[1] Huang, DeepAL: Deep Active Learning in Python

**Deanonymize Review:**

no

**Final Rating After The Rebuttal:**

4: Weak Accept

**Justification Of The Final Rating:**

I would like to thank the authors for the rebuttal. I update my rating as "weak accept" since I believe using meta-learning for selecting images to be labeled is quite interesting. I think the medical imaging community can benefit and build on top of this paper.

**Paper Type:**

methodological development

**Questions To Address In The Rebuttal:**

It would be helpful if the authors can provide answers to some of the following questions.
1 - How does the method perform on different datasets?
2 - How does the method perform on different held-out datasets?
3 - How does the method compare with the other active learning-based methods in the literature?


**Special Issue:**

no

---

### Official Review · Reviewer_79Ru · 2022-01-24

**Confidence:** 3
**Preliminary Rating:** 4
**Recommendation:** Poster

**Summary:**

The authors propose a selective labeling approach via meta-learning to address limited label cases for medical imaging applications. The proposed work consists of: (a) a deep learning model that selects which images to label based on its image embeddings obtained from pre-training in a contrastive setup, and (b) a non-parametric classifier that classifies unseen images using cosine similarity. They evaluate it on chest X-ray interpretation of seen and unseen medical conditions. They show that it learns a good selection strategy and performs better than compared baselines.


**Strengths:**

1. It is a well-written article, and the methods section is easy to follow.
2. The earlier works and their shortcomings, and how the proposed work differs from them are clearly presented.
3. The experiments and appropriate comparisons with earlier works are well done.
4. The additional analysis presented in section 4.2 to study how the proposed strategy selects X-rays differently to compared methods (clustering and random selection strategies) is interesting and is useful for the readers.


**Weaknesses:**

P1. Can the authors clarify how the 10000, 1000, and 2000 tasks for meta-training, meta-validation, and meta-testing are constructed from just 8 medical conditions? What does each task consist of?

P2. Can the authors provide the intuition behind computing the difference between cosine similarities for a new example x^{Q} from Q on page 4 bottom. Where is this difference value used later?

P3. In optimization, can the authors explain how the AUROC score is computed for each query example?

P4. How do we interpret the Wasserstein distance in Figure 3? What is the expected value for each comparison, for example, should we expect a high or low value for MedSelect vs Random compared to Clustering vs Random? Similarly, should the variance of this distance be higher for MedSelect vs Random compared to other combinations?


**Deanonymize Review:**

no

**Final Rating After The Rebuttal:**

4: Weak Accept

**Justification Of The Final Rating:**

The authors have addressed most of the concerns raised and have provided valid justifications or clarifications.
This is an interesting idea that is worth discussing at the conference.
I have no further comments.

**Paper Type:**

methodological development

**Questions To Address In The Rebuttal:**

The authors can comment/clarify the points mentioned in the weakness section.

1. The task formulation can benefit from more details on how each task in these 10000+ tasks is constructed [P1].
2. Some clarification on the equations and AUROC computation [P2, P3] and Wasserstein distance interpretation [P4] would be great.

**Special Issue:**

no

---

### Official Review · Reviewer_vaGC · 2022-01-26

**Confidence:** 4
**Preliminary Rating:** 3
**Recommendation:** Poster

**Summary:**

The paper presents a method for active learning in the context of medical image classification, which is based on meta-learning. In this method, an LSTM-based selector is given a set of unlabelled images as input, and outputs a probability distribution over the images which is then used via sampling to select images for labeling. In meta-training, the labels of an oracle (the expert) are used to define a simple prototype-based binary classifier where the prototype of each class (normal or condition) is defined as the mean representation of selected examples in this class. The selector's parameters are optimized using the REINFORCE algorithm with the classifier's AUC as reward. The proposed method gives improved results compared to random sampling, cluster-based sampling and sampling based on clinical data, especially when the number of selected samples is low.

**Strengths:**

* While several works have explored meta-learning approaches for active learning, the application of such technique to medical image classification has been more limited. Moreover, the combination of meta-learning with contrastive representative learning, proposed in the paper, seems novel.

* The experiments and analysis of results, which is structured around relevant questions, is interesting. Results show the proposed method to outperform three baselines based on random sampling, clustering, and clinical data.

* The experimental validation and discussion consider several challenges in active learning including potential sampling bias and generalization to unseen medical conditions.


**Weaknesses:**

* The literature review could be improved to better motivate the use of meta-learning and single-step labeling, and to clarify the novel contributions of the work with respect to existing approaches as (Contardo et al., 2017). As mentioned in (Liu et al, 2021), meta-learning is a well-known technique for active learning. The literature review could also be expanded to include other works on active learning for medical imaging, for example (Shi et al., 2019).

* Experiments compare the method against simple baselines, and not state-of-art approaches for active learning. As I understand, all baselines use the same simple prototype-based classifier; a stronger evaluation should also include approaches that do not have this restriction.

* The presentation of method is sometimes hard to follow and could benefit from a pseudo-code algorithm.

Liu, P., Wang, L., He, G. and Zhao, L., 2021. A Survey on Active Deep Learning: From Model-driven to Data-driven. arXiv preprint arXiv:2101.09933.

Shi, X., Dou, Q., Xue, C., Qin, J., Chen, H. and Heng, P.A., 2019, October. An active learning approach for reducing annotation cost in skin lesion analysis. In International Workshop on Machine Learning in Medical Imaging (pp. 628-636). Springer, Cham.



**Deanonymize Review:**

no

**Detailed Comments:**

Other comments:

* In the Data section, it is mentioned that 70% of the data is randomly selected for training. Where any preprocessing steps like duplicate removal applied to prevent data leakage? Also, it would be good if the splits are stratified to have a fair representation of important groups.

* In Figure 1, it would be useful to include the pretrained encoder to clarify that the selector and classifier are operating on the embeddings and not the actual images.

* Is the LSTM-based selector sensitive to the ordering of images in U? Are any techniques like shuffling used to remove make the model invariant to the ordering? Would permutation invariant models like transformers be more suitable for this selection?

* In the experiments, why not compare the method against a broader range of active learning techniques, for instance technique based on entropy, uncertainty estimation, ensembling, reinforcement learning, etc.?

* In the results, why is there a sampling bias toward frontal X-rays and younger population? Normally, active learning methods (e.g., approaches based on core sets) seek to have a diversified selection of examples.


**Final Rating After The Rebuttal:**

3: Borderline

**Justification Of The Final Rating:**

I appreciate the authors' response regarding using an LSTM instead of a Transformer. Adding the Transformer results would make the paper stronger. However, I feel that responses to other comments were not sufficient. The lack of strong comparison baselines in the experiments, which was pointed out by another reviewer, was not addressed. Also, the paper was not improved to clarify the methodology.

**Paper Type:**

methodological development

**Questions To Address In The Rebuttal:**

Better motivate the choice of the single-step meta-learning setup and clarify contributions with respect to the literature. Compare the method against a broader range of approaches for active learning. Explain the bias of the model toward frontal X-rays of the younger population.

**Special Issue:**

no

---

### Meta-Review · Area_Chair_qFEm · 2022-02-16

**Recommendation:** Accept (Oral)
**Confidence:** 4

**Metareview:**

The reviewers are very positive about the paper and are of the opinion that this would be a very interesting read to the medical imaging community. I find the rebuttal has addressed most concerns and the reviewers are satisfied with the responses.

---

### Decision · Program_Chairs · 2022-02-28

Accept